# Distinctive Phenotype of Multisystem Inflammatory Syndrome in Children Associated with SARS-CoV-2 According to Patients’ Age: A Monocentric Experience

**DOI:** 10.3390/children9040468

**Published:** 2022-03-27

**Authors:** Antonietta Giannattasio, Francesca Orlando, Carolina D’Anna, Stefania Muzzica, Francesca Angrisani, Sabrina Acierno, Francesca Paciello, Fabio Savoia, Maria Tardi, Angela Mauro, Luigi Martemucci, Vincenzo Tipo

**Affiliations:** 1Pediatric Emergency and Short Stay Unit, Santobono-Pausilipon Children’s Hospital, 80121 Naples, Italy; dannacarol@hotmail.it (C.D.); stefaniamuzzica@hotmail.it (S.M.); francesca.angrisani92@gmail.com (F.A.); sabrina.acierno@gmail.com (S.A.); angela.mauro84@gmail.com (A.M.); enzotipo@libero.it (V.T.); 2Pediatric Department, Santobono-Pausilipon Children’s Hospital, 80122 Naples, Italy; francesca.orlando8@gmail.com (F.O.); paciellofrancesca90@gmail.com (F.P.); tardi.maria@libero.it (M.T.); l.martemucci230@gmail.com (L.M.); 3Childhood Cancer Registry of Campania, Santobono-Pausilipon Children’s Hospital, 80129 Naples, Italy; fabsavoia@gmail.com

**Keywords:** MIS-C, clinical features, cardiac dysfunction

## Abstract

Background: Multisystem inflammatory syndrome in children (MIS-C) is a disease temporally related to severe acute respiratory syndrome coronavirus 2 (SARS-CoV-2) and it is characterized by fever, conjunctival injections, rash, gastrointestinal symptoms, and cardiovascular complications. We evaluated the clinical presentation, laboratory findings, imaging features, therapeutic interventions, and hospital course of a monocentric cohort, and we analyzed these findings according to two age groups. Methods: Patients with MIS-C admitted to a Tertiary Care Pediatric Hospital from November 2020 to November 2021 were considered for the enrollment. Results: Overall, 35 consecutive patients were included. Most of the children did not require intensive care unit at the admission. The clinical presentation of MIS-C slightly differs according to age groups. Mucocutaneus involvement was more frequent in younger patients, while abdominal symptoms were present in 54% of patients aged less than 5 years and in 95% of patients aged more than 5 years (*p* < 0.05). In addition, the number of cases with troponin above the normal reference value was significantly higher in older patients (77%) compared to younger cases (15%) (*p* < 0.01). Conclusions: MIS-C is a new emerging condition and represents a challenge to pediatricians due to the severity of presentation. Further studies to better characterize the long-term outcome of MIS-C patients are mandatory.

## 1. Introduction

Coronavirus disease 2019 (COVID-19), caused by severe acute respiratory syndrome coronavirus 2 (SARS-CoV-2), was described as asymptomatic or mildly symptomatic in more than 80% of children, causing few pediatric hospitalizations and minimal mortality [1,2,3]. Since April 2020, several cases of multisystem inflammatory syndrome in children (MIS-C), a syndrome temporally associated with SARS-CoV-2, have been reported [4,5,6]. The case definition of MIS-C has been published by the U.K. Royal College of Paediatrics and Child Health (RCPCH), the U.S. Center for Disease Control and Prevention (CDC), and the World Health Organization (WHO) [7,8]. MIS-C partially overlapping with features of Kawasaki disease (KD), an acute vasculitis of medium vessel that led to coronary artery aneurysms in ≈25% of untreated cases [9]. However, some differences are present: KD is common in children <5 years, while MIS-C prevalently involves children >5 years of age; gastrointestinal symptoms are more common in MIS-C than in KD; platelets in MIS-C are often low; cardiovascular complications are mostly present as coronary involvement in KD but as myocarditis in MIS-C [9,10,11]. Based on these considerations, we evaluated the clinical presentation, laboratory findings, imaging features, therapeutic interventions, and hospital course of patients with MIS-C to point out differences according to age (less or more than 5 years).

## 2. Materials and Methods

All patients up to 14 years of age sequentially hospitalized at the Santobono-Pausilipon Children’s Hospital with a diagnosis of MIS-C from 1 November 2020 to 1 November 2021 were prospectively enrolled. This hospital is a Tertiary Care Pediatric Hospital in the Campania region (Southern Italy), and it has been identified as the Regional Spoke for the diagnosis of children’s SARS-CoV-2 infection. The diagnosis of MIS-C was performed according to the CDC criteria [12]. Data on demographics, exposure history, medical history, clinical presentations, laboratory tests, imaging investigations, and treatments were collected. Only patients with a certain diagnosis of MIS-C and complete medical chart reviews were included in the study. Patients with a diagnosis of severe acute SARS-CoV-2 infection or with a typical KD were excluded. Laboratory investigations at admission included a complete blood count, inflammatory markers (C-reactive protein, CRP, procalcitonin, PCT, ferritin, interleukin-6, IL6, and IL-2 receptor, IL2r), kidney and liver functional tests, electrolytes, albumin, troponin, b-type natriuretic peptide (BNP), SARS-CoV-2 serologic assay, and a nasopharyngeal swab for SARS-CoV-2 detection. Blood, stool, and urine cultures, a multiplex nucleic acid amplification test for multiple respiratory pathogens, Epstein–Barr Virus and cytomegalovirus antibodies, and a peripheral blood smear were also performed. All children underwent cardiac imaging and an abdominal ultrasound during the first days of hospitalization. The treatment strategies were supported by the American College of Rheumatology Clinical Guidance and treatment guidance from the Rheumatology Study Group of the Italian Society of Pediatrics [13,14]. Mild-moderate cases were admitted to our regular ward; if patients needed a continuous monitoring, they were admitted to the sub-intensive care unit, while if they needed a mechanical ventilation, vasopressors and/or inotropes, or in the case of hemodynamic instability, they were admitted to the pediatric intensive care unit (PICU).

Medians, interquartile ranges for continuous variables and numbers and percentages for categorical variables were used. Demographic, clinical, and laboratory features were compared between age groups (≤5 years and >5 years). Categorical variables were evaluated with chi-square test and Fisher’s test, as appropriate. Wilcoxon’s rank sum test was used for continuous variables. A *p* value < 0.05, two-tailed, was considered statistically significant. The analysis was performed on StataCorp LLC Stata 13.0 (College Station, TX, USA).

## 3. Results

### 3.1. Characteristics of Patients with MIS-C

We included 35 patients fulfilling the diagnostic criteria for MIS-C (Table 1). In the majority of cases, MIS-C cases occurred about 3 weeks after the peak of daily COVID-19 cases in the Campania region (peak on November 3, 2020, 21.5% of positive nasopharyngeal swab following the peak of SARS-CoV-2 infection; following peaks of SARS-CoV-2 infection in January, February, March, and August 2021). A minority of cases was sporadic and was observed irrespective of SARS-CoV-2 peaks [15]. A SARS-CoV-2 serologic assay revealed IgG positivity in all cases. Only seven (20%) patients had a history of a previous known SARS-CoV-2 infection. No severe acute COVID-19 has been reported. In all cases, no other causes responsible for the clinical picture were found (microbiological, virological, and serological tests for pathogens other than SARS-CoV-2 and the peripheral blood smear were negative in all cases).

All but one (a girl from Southeast Asia) were of Caucasian ethnicity. As for clinical presentation, all patients presented fever. Mucocutaneous involvement was present in 27 (77%) cases, while gastrointestinal symptoms in 26 (74%) children. During hospital stay, 28 (80%) patients presented a cardiac involvement (increased troponin levels and/or echocardiography abnormalities). No patient had coronary artery aneurysm and/or dilatation.

All patients had elevated inflammatory index at baseline. During hospitalization, 33 (97%) children showed elevated D-dimer and 28 (82%) high fibrinogen levels. Ten (29%) patients had a sodium level <132 mmol/L. Hepatic involvement with hypertransaminasemia was detected in 10 (29%) cases. No patient had cholestasis. Hyperamylasemia was detected in 10 (28%) cases. Troponin was elevated in 19 (54%) patients and a high BNP level was found in 60% (21 children) of cases (elevation of both troponin and BNP was found in 15 cases, 43%). No case of acute kidney injury was found. The majority (28 patients, 80%) of patients had abnormalities in abdominal imaging. Abdominal ultrasound showed mesenterial lymphadenitis in 19/28 (68%) patients, terminal ileitis and bowel wall thickening in 17 (61%) cases, and ascites in 16 (57%) patients. The majority of cases presented a combination of these three features. As for cardiac involvement, small changes in cardiac conduction and repolarization were observed in the majority of cases (30/35 patients). The most common ECG alteration was transient first-degree atrio-ventricular (AV) block (18/30 cases, 60%), followed by bradyarrhythmia (16 cases, 53%). The majority of patients (26/35, 74%) had one or more abnormalities in cardiac imaging. The most common abnormal findings were mitral regurgitation (*n* = 22, 85%), pericardial effusion (*n* = 11, 42%), and left ventricle (LV) dysfunction (*n* = 11, 39%). Six (21%) patients showed severe myocarditis. No child had coronary artery abnormalities. When cardiac abnormalities were correlated with the clinical and laboratory presentation of MIS-C patients, we only found a significantly higher ferritin level in patients with abnormal echocardiography (759 ng/mL, 465–1073) compared to nine patients with no cardiac alteration (461 ng/mL, 365–1073) (*p* = 0.02).

All patients started specific treatment within 48 h since hospital admission (Table 1). After an incomplete response to initial therapy with IVIG and corticosteroids, anakinra was administered in five cases with rapid clinical and laboratory improvements.

At hospital discharge, an improvement of inflammatory markers and cardiac indexes was observed in all cases.

### 3.2. Difference According to Age Groups

In order to point out an age-related phenotype of MIS-C, patients were divided into two age groups: Group 1 (≤5 years) and Group 2 (>5 years) (Table 2). As for the clinical presentation of MIS-C, mucocutaneus involvement did not statistically differ between the two groups, while abdominal symptoms were more frequent in Group 1 (7/13, 54%) compared to Group 2 (21/22, 95%; *p* < 0.05). As for laboratory investigation, Group 2 had a significantly lower lymphocyte count (*p* < 0.01) and a high troponin value (*p* < 0.01) compared to Group 1. The number of cases with troponin above the normal reference value was significantly higher in Group 2 (17/22, 77%) compared to Group 1 (2/13, 15%) (*p* < 0.01). The number of patients with hypertransaminasemia and hyperamylasemia did not differ between the two groups. As for abdominal imaging, a significantly higher percentage (14 children, 64%) of patients belonging to Group 2 had inflammation in the ileum and colon compared to Group 1 (three patients, 23%; *p* = 0.02). Other abdominal abnormalities (as ascites, hepatomegaly, and liver steatosis) did not significantly differ between the two groups. Although cardiac abnormal findings were more common in older patients compared to children <5 years, an analysis of the type of cardiac imaging dysfunction revealed no difference between the two groups for mitral regurgitation pericardial effusion, LV dysfunction, and myocarditis.

As for treatment, patients belonging to Group 2 required a high dose of intravenous steroids compared to Group 1 (*p* = 0.04).

### 3.3. Severity of Presentation

Most of the children (65%, 23 patients) were admitted to our regular pediatric ward. Eight (23%) patients were admitted to our sub-intensive care unit while three (9%) cases required PICU observation for 24–48 h because of hemodynamic instability, and they were then transferred to our sub-intensive unit. One (3%) child died a few hours after its admission in the PICU because of cardiac failure and consequent irreversible shock syndrome. No other death occurred in our cohort.

## 4. Discussion

This study provided a clear picture of clinical, laboratory, and imaging features of children with MIS-C according to the age of the patients. We included only patients fulfilling the diagnostic criteria for MIS-C and with a positivity of IgG anti-SARS-CoV-2. Some reports included patients with MIS-C but negative IgG anti-SARS-CoV-2, or those with any suspected inflammatory illness after SARS-CoV-2 infection [16,17]. It is to note that MIS-C is a post-infection phenomenon rather than a condition resulting from an acute SARS-CoV-2 infection [18,19]. The severity of acute COVID-19 is not related to the possibility to develop MIS-C [18,19]. This is confirmed by our results, considering that the majority (80%) of patients had an unknown previous SARS-CoV-2 infection and that the remaining 20% of cases had an asymptomatic or mild acute SARS-CoV-2 infection.

The spectrum of severity of MIS-C in our cohort varied from cases with a prompt response to therapy and a moderate clinical course, to patients with cardiac involvement and sudden cardiac shock with multiorgan failure and death (in one case). The most frequent clinical presentation was with mucocutaneous and gastrointestinal symptoms/signs. This is in line with the published literature [16,20]. However, we found differences according to the patient’s age; gastrointestinal presentation and cardiac involvement were more common in patients aged >5 years than in younger patients. The mortality of MIS-C is estimated to be around 1% based on U.S. data [21]. Its prognosis is mainly related to cardiac damage. In our cohort, the majority of patients showed cardiac abnormalities in the ultrasound, although a severe myocarditis was found in only 6/35 patients. It has been reported that the acute echocardiographic findings appear to be usually reversible in a short-term follow-up, with a normalization of systolic function and a full recovery of cardiac damage [22,23,24]. Clearly, further studies including large samples and a long follow-up are needed before a definitive statement can be made.

Compared to the majority of reports, PICU admission was low and very few patients required inotropes [16,25]. This difference may be due, at least in part, to the organization of the Italian Health System, which guarantees a prompt access to the Pediatric Emergency Department, and to a short duration of symptoms before hospital admission. Furthermore, in all cases of our cohort, medications were started within 48 h from admission so that the inflammatory cascade of MIS-C was promptly interrupted.

As for treatments, almost all cases (89%) received IVIG, with the addition of corticosteroids in case of the failure of IVIG therapy or of an onset with cardiac involvement. Biological treatment with anakinra was mandatory in patients unresponsive to IVIG and corticosteroids. No other biological modifying medication (such as infliximab or tocilizumab) was used in our cohort.

A point of strength of our study is the inclusion of patients with a definitive diagnosis of MIS-C. Capone et al. described 33 patients with MIS-C, of whom 21 met the complete criteria for KD [20]. Furthermore, in this study, patients with acute COVID-19 were considered together with MIS-C patients [20]. In other studies, about 40% of MIS-C patients also had a diagnosis of KD [26,27]. In our cohort, we excluded typical KD in order to better characterize MIS-C.

A main limitation of our study was the single-center experience. This limits the number of included cases. However, compared with other single-center studies [28], the number of studied children can be considered representative, while also taking into account the relatively low frequency of MIS-C. Furthermore, a single-center experience allowed us to have a homogeneous diagnosis, management, and treatment protocol.

In conclusion, the management of MIS-C requires a multidisciplinary approach. Cardiac dysfunction is the most serious complication, and it seems to be prevalent in older children. However, in a short-term follow-up, a good functional recovery was observed. Further studies to better characterize the long-term outcome of MIS-C patients are mandatory.

## Figures and Tables

**Table 1 children-09-00468-t001:** Demographic, clinical, laboratory, and imaging characteristics and treatment in 35 patients with MIS-C.

Variables	Value
Males ^+^	18 (51%)
Age in years *	7 (4–10)
Underlying medical conditions (excluding obesity) ^+^	1 (3%) **
Obesity ^+^	7 (20%)
Duration of symptoms before hospital admission in days *	3 (3–5)
Clinical presentation:	
- Fever ^+^	35 (100%)
- Conjunctivitis	24 (69%)
- Rash	17 (49%)
- Cheilitis	12 (34%)
- Oral changes	17 (49%)
- Cervical lymphadenopathy	3 (9%)
- Abdominal pain	21 (60%)
- Diarrhea	17 (49%)
- Meningism	5 (14%)
Length of hospitalization in days *	12 (8–15)
Laboratory values: *	
- WBC count/µL (×1000)	11.7 (7.9–15.7)
- Lymphocyte count/µL (×1000)	1.2 (1.0–2.1)
- Platelet count/µL (×1000)	206 (162–252)
- CRP (mg/L) (ref: <0.5)	145 (106–211)
- PCT (ng/mL) (ref: <0.05)	2.4 (0.7–11.2)
- Ferritin (times upper the normal value)	8.3 (5.1–17.2)
- Fibrinogen (mg/dL) (ref: 180–400)	567 (414–748)
- D-dimer (ng/mL) (ref: <270)	1084 (806–1844)
- AST (IU/L) (ref: <58)	35 (21–58)
- ALT (IU/L) (ref: <40)	28 (16–54)
- Amylase (U/L) (ref: <80)	35 (25–51)
- Albumin (g/dL)	3.4 (3.3–3.7)
- Lowest albumin (g/dL)	2.8 (2.1–3.7)
- LDH (U/L) (ref: <550)	460 (361–567)
- Troponin (ng/L) (ref: <14)	21 (8–51)
- BNP (pg/mL) (ref: <100)	125 (58–745)
- IL2r (IU) (ref: <710)	4031 (2197–8772)
- IL6 (pg/mL) (ref: <5)	45 (11–159)
Abnormal findings in imaging: ^+^	
- Abdominal ultrasonography	28 (80%)
- Electrocardiograms	31 (89%)
- Echocardiography	26 (74%)
Number of organs/systems involved: ^+^	
- 2	1 (3%)
- 3	6 (17%)
- ≥4	28 (80%)
Medications for MIS-C: ^+^	
- Antimicrobials	32 (91%)
- IVIG	31 (89%)
- Bolus of intravenous steroids	17 (49%)
- Low dose of intravenous steroids (without bolus)	29 (83%)
- Anakinra	5 (14%)
- Vasoactive drugs	1 (3%)
- Enoxaparin	10 (29%)
- Aspirin	21 (60%)

WBC, white blood cells; AST, aspartate aminotransferase; ALT, alanine aminotransferase; LDH, lactate dehydrogenase; CRP, C-reactive protein; PCT, procalcitonin; BNP, b-type natriuretic peptide; Il2r, interleukin-2 receptor; IL6, interleukin-6; IVIG: intravenous immunoglobulins.; ^+^ number (%); * median (interquartile range); ** Pulmonary artery stenosis.

**Table 2 children-09-00468-t002:** Demographic, clinical, laboratory, and imaging characteristics and treatment in patients with MIS-C divided in two age groups.

Parameters	Group 1: ≤5 Years (*n* = 13)	Group 2: >5 Years (*n* = 22)	*p*
Males ^+^	8 (62%)	10 (45%)	ns
Obesity ^+^	0	6 (27%)	ns
Clinical presentation: ^+^			
- Fever	13 (100%)	22 (100%)	
- Conjunctivitis	11 (85%)	13 (59%)	ns
- Rash	9 (69%)	8 (36%)	ns
- Cheilitis	4 (31%)	8 (36%)	ns
- Oral changes	8 (62%)	9 (41%)	ns
- Cervical lymphadenopathy	1 (8%)	2 (9%)	ns
- Abdominal pain	4 (31%)	17 (77%)	0.01
- Diarrhea	5 (38%)	12 (55%)	ns
- Meningism	6 (46%)	6 (27%)	ns
Length of hospitalization in days *	8 (6–10)	13 (12–16)	0.04
Laboratory values: *			
- WBC count/µL (×1000)	13.1 (11.7–15.8)	10.2 (7.9–15.5)	ns
- Lymphocyte count/µL (×1000)	2.2 (1.9–3.2)	1.0 (0.8–1.3)	<0.01
- Platelet count/µL (×1000)	225 (170–252)	198 (151–242)	ns
- CRP (mg/L)	144 (115–179)	146 (83–216)	ns
- PCT (ng/mL) (ref: <0.05)	1.4 (0.7–10.9)	2.4 (0.8–11.2)	ns
- Ferritin highest value (times upper the normal value)	7.9 (5.0–11.6)	9.3 (6.4–19.4)	
- Fibrinogen (mg/dL) (ref: 180–400)	529 (399–720)	597 (461–765)	ns
- D-dimer (ng/mL) (ref: <270)	928 (658–1636)	1116 (883–1844)	ns
- AST (IU/L) (ref: <58)	28 (21–51)	38 (20–58)	ns
- ALT (IU/L) (ref: <40)	18 (15–50)	31 (17–54)	ns
- Amylase (U/L) (ref: <80)	30 (22–37)	42 (27–58)	ns
- Albumin (g/dL)	3.4 (3.0–3.7)	3.4 (3.0–3.7)	ns
- LDH (U/L) (ref: <550)	515 (495–570)	395 (345–542)	ns
- Troponin (ng/L) (ref: <14)	9 (5–13)	27 (21–53)	<0.01
- BNP (pg/mL) (ref: <100)	151 (75–620)	123 (57–745)	ns
- IL2r (IU) (ref: <710)	6148 (2728–9681)	4031 (1258–9681)	ns
- IL6 (pg/mL) (ref: <5)	122 (102–297)	29 (8–70)	ns
Abnormal findings in imaging: ^+^			
- Abdominal ultrasonography	8 (67%)	20 (91%)	0.07
- Electrocardiograms	11 (85%)	20 (91%)	ns
- Echocardiography	8 (62%)	18 (82%)	ns
Number of organs/systems involved: +			
- 2	0	1 (5%)	
- 3	2 (15%)	4 (18%)	ns
- ≥4	11 (85%)	17 (77%)	
Medications for MIS-C: +			
- Antimicrobials	12 (92%)	20 (91%)	ns
- IVIG	11 (85%)	20 (91%)	ns
- Bolus of intravenous steroids	3 (23%)	14 (64%)	0.04
- Low dose of intravenous steroids (without bolus)	9 (69%)	20 (91%)	ns
- Anakinra	0	5 (23%)	ns
- Vasoactive drugs	1 (8%)	0	ns
- Enoxaparin	2 (15%)	8 (36%)	ns
- Aspirin	8 (62%)	13 (59%)	ns

WBC, white blood cells; AST, aspartate aminotransferase; ALT, alanine aminotransferase; LDH, lactate dehydrogenase; CRP, C-reactive protein; PCT, procalcitonin; BNP, b-type natriuretic peptide; Il2r, interleukin-2 receptor; IL6, interleukin-6; IVIG: intravenous immunoglobulins; ns: not significant. ^+^ number (%); * median (interquartile range).

## Data Availability

Not applicable.

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
