# Peer review of "Distinctive Phenotype of Multisystem Inflammatory Syndrome in Children Associated with SARS-CoV-2 According to Patients’ Age: A Monocentric Experience"

_children, 2022, doi:10.3390/children9040468_

Round 1

Reviewer 1 Report

This is single center study reviewing clinical experience with MIS-C. The novelty is in defining the difference between the "KD age group" and older children which is interesting and reflect what we see in clinic. 

  1. In results - for mucucutaneous symptoms, there is no significant difference and descriptive difference is not impressive, better phrase as - no difference, for scientific soundness
  2. Could the authors add echo findings? could they test for any association between cardiac involvement and the presenting symptoms / degree of illness?
  3. Death in MIS-C population is very rare, could the authors expand on this particular case?

Reviewer 2 Report

This is an important study in that it contributes further information about MIS-C and specifically highlights some of the differences in younger vs older patients. Some differences in clinical presentation were noted, though not many. The study would be strengthened by providing more detail in general. Symptoms are broadly categorized and minimally described and the manuscript should go into more detail (particularly because the tables presented break down the broad categories into narrower ones, but the text quotes combinations of this data). There is an error in the discussion regarding which group presented more frequently with GI symptoms, at least based on the data provided.  Additionally, substantial editing is needed for grammatical/English language issues.

Reviewer 3 Report

Undoubted, MIS-C is an emerging condition in children, deserving the attention of paediatricians and scientists. Collecting adequate qualitatively and quantitively data is crucial for understanding the clinical course, pathogenetic mechanisms, the appropriate therapeutic approach and prognosis. Furthermore, the age factor is an essential determinant in many conditions in childhood. In this respect, the Antonietta Giannattasio et al. study is a hot topic. The study has many advantages highlighted in the title and detail in the main text. An additional advantage is a complete laboratory (including interleukins) and imaging evaluation of the included cases.

However, there are some comments and issues to be addressed:

The title needs to be reconsidered in part concerning the cohort scale – “large”.

The cases are collected for one year period - November 2020 to November 2021, but only the November 2020 peak of acute COVID-19 cases is mentioned in the results (lines 84-85). Is there the same latent period between the subsequent peaks and the MIS-C cases? Are all the cases clustered in specific periods or appears sporadically as well? 

When defining the acute COVID-19 peak, it is not clarified if 21.5% of positive nasopharyngeal swabs are taken from all age groups, particularly in children.

Material and methods are mentioned - Blood-, stool- and 64 urine-cultures, multiplex nucleic acid amplification test for multiple respiratory pathogens, Epstain Barr Virus and cytomegalovirus antibodies, peripheral blood smear, but in the Results, there is no data included. Are all the microbiological, virological and serological tests negative?

Abnormal Imaging findings are reported. However, it is not described what abnormalities are observed – abdominal ultrasound – ascites, mesenterial lymphadenitis or others? Echocardiography – myocarditis, pericarditis? Is there any differences in the abnormal imaging findings qualitatively, not only quantitively as +number (%). What are the main ECG pathological findings in the cohort and the age subgroups, respectively? 

What age is the case with pulmonary artery stenosis, and is there a connection between this comorbidity and the severity of presentation. 

What is the ethnicity of the included children? It is known to connect with the severity of presentation, which is one of the study evaluations.

Table 1 – spaces are missing between words – Cervicallymphoadenopathy, Abdominalpain, whitebloodcells ….

In table 2 – please check the D-dimer and CRP values. There is a big discrepancy in values between table 1, table 2 and the statements in the main text. Check the IL2r and IL6 in the two groups; some values are missing.

According to the literature, currently, the children surviving after MIS-C demonstrate fully recovering. However, is there follow-up data, particularly in children with cardiac involvement?

In the abstract, it is stated that “…abdominal symptoms were present in 54% of patients aged less than 5 years and in 95% of patients ages more than 5 years (p<0.05).” 
In the discussion, it is pointed different observation “However, we found differences according to the age’s patient, being gastrointestinal presentation prevalent in younger children and cardiac involvement more common in patients aged >5 years.”(lines 153-155). Please clarify which statement is correct?

Conclusion and practical implications are missing in the paper. 

Round 2

Reviewer 3 Report

The authors significantly improved the manuscript, and many of the comments have been addressed.

However, there are still some issues to be clarified.

In the added information for the abdominal ultrasound imaging is evident that the most common feature of MIS-C – ascites is not mentioned. Are there cases with ascites in this cohort? The same question for the mesenterial lymphadenitis – another very typical feature for the syndrome, especially in cases with leading abdominal symptoms.

Related to the ascites, it makes an impression that the albumin level of the enrolled children is within the normal ranges (3.4 (3.3-3.7), not reduced as expected. Do you have an explanation or interpretation for those findings concerning the data in the literature?

Another comment is the lack of ethical approval and informed assent/consent information from the parents/enrolled children.
